# A closer look at the timecourse of mind wandering: Pupillary responses and behaviour

**Claudia Pelagatti**[1], **Paola Binda**[2]*, **Manila Vannucci**[1]*

**1** Department of NEUROFARBA-Section of Psychology, University of Florence, Florence, Italy,
**2** Department of Translational Research and New Technologies in Medicine and Surgery, University of Pisa, Pisa, Italy

* paola.binda@unipi.it (PB); manila.vannucci@psico.unifi.it (MV)

## Abstract

Mind wandering (MW) refers to the shift of attention away from a primary task and/or external environment towards thoughts unrelated to the task. Recent evidence has shown that pupillometry can be used as an objective marker of the onset and maintenance of externally-driven MW episodes. In the present study we aimed to further investigate pupillary changes associated with the onset and duration of self-reported MW episodes. We used a modified version of the joint behavioural-pupillometry paradigm we recently introduced. Participants were asked to perform a monotonous vigilance task which was intermixed with task-irrelevant cue-phrases (visually presented verbal cues); they were instructed to interrupt the task whenever a thought came to mind (self-caught method) and to indicate the trigger of their thought, if any. We found systematic pupil dilation after the presentation of verbal cues reported to have triggered MW, compared with other verbal cues presented during a supposedly on-task period (i.e., the period immediately following the resuming of the task after a self-caught interruption and MW report). These results confirm that pupil diameter is sensitive to the changes associated with the onset of MW and its unfolding over time. Moreover, by computing the latency between the trigger presentation and the task interruption (self-catch), we could also estimate the duration of MW episodes triggered by verbal cues. However, a high variability was found, implying very large inter-event variability, which could not be explained by any of the MW properties we acquired (including: temporal focus, specificity, emotional valence). Our behavioural and pupillometry findings stress the need for objective measures about the temporal unfolding of MW (while most studies focus on arbitrary time-window preceding self-reports of MW).

## Introduction

In our daily lives, we may notice that our attention drifts away from the ongoing task and external environment, and our mind starts wandering elsewhere, towards task-unrelated private thoughts and feelings such as memories, future plans, current concerns. This *"shift in the focus of attention away from the here and now towards one's private thoughts and feelings"* [1, p. 818] is referred to as mind wandering [hereafter MW; for review, 2]. This shifting away is in

**Data Availability Statement:** The data are available in Zenodo, at the following doi and URL: https://doi.org/10.5281/zenodo.3673044.

**Funding:** This project has received funding from the European Research Council (ERC) under the

European Union's Horizon 2020 research and innovation programme (grant agreement to P.B., No 801715 - PUPILTRAITS). The funders had no role in study design, data collection and analysis, decision to publish, or preparation of the manuscript.

**Competing interests:** The authors have declared that no competing interests exist.

most cases spontaneous, although evidence has been reported that it might also occur intentionally [3].

During the past two decades, MW has received considerable scientific interest, with a steep rise of publication numbers in cognitive and clinical psychology and in neuroscience. Over the last years, two important and related issues, conceptually and methodologically relevant, have been raised in the field. The first one refers to the question of the conceptual and operational definition of MW: although most researchers would agree with the general definition of MW reported above, in the studies published over the last decades different operational definitions of MW have been provided, including task-unrelated thought, off-task thought, stimulus-independent thought/perceptually decoupled thought, stimulus independent and task-unrelated thought, unintentional thought, self-generated thought, resulting in some conceptual confusion in the literature on this topic. This heterogeneity may also have contributed to the variability of findings reported in the literature on the behavioral and psychophysiological correlates of the experience of MW (e.g. pupillometry, eye-movements, evoked potentials). Although different theoretical approaches and solutions have been recently suggested to deal with this issue [see for a discussion, 4–6] researchers agree with the necessity of providing an explicit definition of the specific experience investigated in each study, to avoid lumping together the vastly different experiences and attentional states described in the studies under the umbrella term of MW.

The second, related, issue refers to the investigation of the dynamics of MW, that is how MW arises and unfolds over time. This issue has been addressed in two theoretical frameworks, the dynamic framework [7] and the process-occurrence framework [8]. The term "mind wandering" implicitly refers to a dynamic mental experience, and to a specific one, that is the "wandering" of our mind, where "*to wander means to "move hither and thither without fixed course or certain aim*" [7, p.2], thereby suggesting that a key-feature of MW is the freedom of movement in thought, that is thoughts can freely move, from one to the next and they can arise with little constraints (i.e. there is no overarching purpose or direction to a person thinking, although there might be some connection between thoughts). Only over the last few years, some studies have started investigating this dynamic dimension of MW, introducing measure of the degree of freedom of movement in thought (i.e., level of constraints on thought as it unfolds over time) [9] and examining its relationship with the other, content-based dimensions, as task-unrelatedness and stimulus-independence [10].

The issue of the dynamic of MW is also addressed, in a more temporal-based perspective, in the process-occurrence framework, proposed by Smallwood [8]. According to this proposal, any comprehensive account of MW is expected to explain *when* and *why* MW occurs, that is which processes and events control and prompt the initial occurrence of MW (onset) and *how* MW unfolds over time, that is which processes sustain MW over time (maintenance). In order to understand how the mind wanders, we need to identify and distinguish between the onset (the so-called process of ignition, see 8) and maintenance.

Indeed, the possibility of empirically addressing this issue is strongly related to the identification of the triggers of MW: MW states need to be causally linked to preceding events in order to study the onset of these experiences and therefore to track their timecourse. In this regard, for a long time MW has been considered as completely stimulus-independent [11] and self-generated mental activity [8]. As already discussed in previous studies [12,13] methodological factors may have contributed to this conceptualization, since most MW research has been using go/no-go tasks with no meaningful and semantically rich cues (in the form of words or images), and participants have not been asked to report if anything trigger their thoughts, thereby allowing for detecting the occurrence of MW but not for identifying the potential triggers of MW experiences.

Over the last years, new experimental paradigms have been introduced, which allow the investigation of the potential contribution of internal (e.g. thoughts, feeling) and external triggers (e.g., environmental stimuli) to MW [12–20]. Globally, the results of these studies demonstrate that the majority of MW is reported to have a clear identifiable trigger, that means that MW might be also stimulus-dependent, in the sense that an internal or external stimulus might induce a shift of our attention [see for discussion, 21]. Moreover, these studies have shown that both task-irrelevant [12,13, 18–20] and task-relevant external stimuli [14–17] might indeed act as triggers for MW.

At the methodological level, an important contribution in investigating the role of task-irrelevant external stimuli in triggering MW has been provided by the vigilance task with cue-words developed by Schlagman and Kvavilashvili [22], and extensively used by different independent research groups to study different kinds of spontaneous cognitions (e,g., involuntary autobiographical memories, mind wandering, involuntary future thoughts) [see for a review and discussion, 21].

In this paradigm, participants are presented with a long sequence of trials of mostly horizontal lines and have to detect an infrequent target (i.e., vertical lines). Participants are also exposed to cue-phrases presented in the center of the screen (e.g., 'missed opportunity' or 'wall mirror'), which they are told are irrelevant to the task. The experience of MW can be assessed during the task by using the self-caught method (i.e., instructing participants to stop the task every time they catch their mind not on-task/mind wandering) or the probe-caught method (i.e., interrupting participants during the task and asking them about their experience just immediately prior to the probe). With both methods, participants are asked to briefly describe their thoughts and to specify the trigger, if any.

The results obtained with this task revealed that (i) the majority of recorded MW episodes are reported as being triggered by the task-irrelevant verbal cues, (ii) the exposure to verbal cues may affect the temporal orientation of MW, increasing the proportion of past-oriented MW [12, 20]. As explained by Plimpton et al [20] it is likely that the presence of meaningful, although task-irrelevant, verbal cues might stimulate bottom-up retrieval processes, resulting in the recall of past memories.

Similar findings have been reported in studies investigating the effects of task-relevant stimuli on MW [14–17]. For example, a very recent study by Faber and D'Mello [14] has shown that during real-world, semantically rich task contexts, as reading an instructional task and watching a video, approximately half of the MW episodes (assessed by self-caught method) were triggered from the task-relevant stimulus and that past-oriented MW were more likely to be triggered from the stimulus, arising from "*spontaneous associations with the stimulus*" [14, p.1], than future-oriented and introspective thoughts.

However, one of the limitations of all these studies is that MW triggers are identified by introspective measures, i.e. asking participants to report what had triggered each MW event. In addition, the use of self-report measures does not allow the monitoring and tracking of the dynamics of MW, such as the changes in the attentional state associated with the onset and maintenance of MW.

In a first attempt to overcome these limitations, we combined self-report measures of MW and pupillometry [13]. Participants performed the vigilance task with task-irrelevant cue-words described above, and MW was assessed by using the probe-caught method–interrupting the task and asking participants to report their thoughts and what had triggered them. In the study we focused on spontaneous MW, and we operationalized MW as thoughts that were unrelated to both the task and the sensory environment; these thoughts could have no trigger, or they might be triggered by internal or external stimuli, which participants identified in their reports. In the categorization of attentional states that we used [see also, for a similar

categorization, 23, 24], we distinguished MW from on-task, external distractions, task-related interferences and blank-mind reports.

The results showed significantly larger pupil dilation following verbal cues reported as triggers of a MW episode compared to the control conditions. The pupil dilation increased over time, suggesting that a change in pupil diameter follows the onset of MW and accompanies its unfolding and maintenance over time. Given the well-known association between pupil dilation and cognitive or emotional load [25,26], we explained the pupil dilation observed after the onset of a MW episode in terms of the increased cognitive and emotional processing associated with the onset of MW and its unfolding, compared to the monotonous vigilance task.

In the present study, we aimed to replicate these findings within a similar paradigm (vigilance task with verbal cues) but using the self-caught rather than probe-caught procedure to assess the occurrence of spontaneous MW episodes (operationally defined in the same way as in our previous pupillometry study).

Although both procedures are considered valid tools for assessing the occurrence of MW, increasing evidence suggest that they may capture distinct episodes of MW and they might provide complementary information about it, each one showing advantages and limitations [see for a recent discussion on this topic, 27].

The self-caught procedure requires participants to monitor their attentional states and the contents of their mind during a task in order to be able to report a MW episode. Since several studies have shown that people are only intermittently aware of their internal state [see for a review 28], only a portion of MW might be reported, and instances of MW that do not reach the level might be missed. With probe-caught method participants are intermittently and pseudo-randomly interrupted and probed with questions regarding the contents of their experience, so that high levels of monitoring are imposed by the questions and participants can report also MW episodes they were not aware of immediately prior to the question (unaware MW).

Both similarities and differences in the outcomes of self and probe-caught MW have been reported. Globally speaking, studies have shown a more consistent pattern of association between probe-caught MW and performance at the ongoing task [see for a review on 145 studies, 29] compared to self-caught method. For example, studies on MW during reading have shown that reading comprehension is negatively correlated with probe-caught MW but not with self-caught MW [e.g., 30,31]. However, the negative association between interest and motivation and MW has been reported for both self [27] and probe-caught MW [32, 33].

In the present study, we aimed to use the self-caught procedure to extend and complement our previous findings on pupillometry correlates of the temporal dynamics of MW. In line with the results obtained with probe-caught method, we expected to find in pupil diameter an index of MW, and specifically a marker of the higher mental load associated with the production and maintenance of MW, compared to the visual processing of vertical and horizontal bars and target detection (on-task state). For this reason, we hypothesized pupil dilation after the presentation of verbal cues reported to have triggered MW, compared with other verbal cues presented after the resume of the task (on-task state).

Moreover, the combination of the vigilance task with cue-words and self-caught method also allowed for addressing an additional question, that we could not address in the previous study with the probe-caught procedure, about the duration of MW. Specifically, for those MW episodes that participants reported as being triggered by verbal cues shown on the screen, we could calculate the latency data or retrieval times, that is the time-interval between the presentation of the trigger and the MW report. This interval reflects the time needed for the arising of thoughts and the awareness of them. In the research field of MW, with a few exceptions [34–36] the question of the potential duration of MW has not been empirically addressed.

Indeed, several studies have examined measures associated with MW states (i.e., reaction time variability, eye movements, fMRI BOLD signal) by using arbitrary time-windows length before self-reports of MW states, assuming that MW episodes occurred precisely in those windows and lasted for that period of time. Here we used participants' self-reports to directly measure the latency of MW episodes triggered by cue-words; we also analyzed the association between latency and some of the main phenomenological properties of MW episodes (i.e., temporal focus, specificity, emotional valence), to explore the possibility that these variables regulate the temporal unfolding of MW.

## Materials and methods

### Participants

Twenty-eight undergraduate students from the University of Florence (age range 19–32 years, M = 21.61 years, SD = 3.06 years, 16 females) volunteered to participate in the study. Four were excluded due to non-compliance with task instructions. Thus, the sample used for analysis included 24 participants (age range 19–32 years, M = 21.50 years, SD = 3.26 years, 14 females).

We computed an approximated sample size for a linear-mixed model using the *smpsize_lmm* function in the R package *sjstats* [37] and we selected a sample size (number of observations) that was adequate for a small-intermediate effect size (d = .4). This required 219 total MW episodes; assuming that each participant experiences 10 MW episodes over the course of the experiment, the power analysis recommends the inclusion of at least 22 participants (which we slightly exceeded to account for possible exclusions of invalid pupillometric data).

All participants were Italian native speakers and they had normal or corrected-to-normal vision. Experimental procedures were approved by the regional ethics committee [Comitato Etico Pediatrico Regionale—Azienda Ospedaliero-Universitaria Meyer—Firenze (FI)] and are in line with the declaration of Helsinki.

### Materials

**Apparatus.**   Subjects sat in front of a monitor screen, with their heads stabilized with a chin rest. Viewing was binocular. Stimuli were generated with the PsychoPhysics Toolbox routines for MATLAB (MATLAB r2010a, The MathWorks) and presented on a LCD colour monitor (Asus MX239H, 51 x 28 cm placed at 57 cm viewing distance) with a resolution of 1920 x 1080 pixels and a refresh rate of 60 Hz, driven by a Macbook Pro Retina (OS X Yosemite, 10.10.5). All stimuli were shown in white (55 cd/m$^2$) against a black background (0.05 cd/m$^2$). Two-dimensional eye position and pupil diameter were monitored binocularly with a CRS LiveTrack system (Cambridge Research Systems) at 30 Hz, using an infrared camera mounted below the screen. Pupil diameter measures were transformed from pixels to millimeters after calibrating the tracker with an artificial 4 mm pupil, positioned at the approximate location of the subjects' left eye. Gaze position data were linearized using a standard 9-point calibration, run prior to each session.

**Vigilance task.**   Participants completed a modified version of the computer-based vigilance task with cue-words, which was originally developed by Schlagman and Kvavilashvili [22]. As detailed in Kvavilashvili et al. [21] this paradigm has been extensively used to investigate different types of spontaneous cognitions, including involuntary autobiographical memories [38–45] involuntary thoughts about the past and future [46–48], and spontaneous mind wandering [12,13,19,20]. The task used in the present study consisted of 1020 trials, presented in fixed order, each lasting 2 s. A white fixation point (0.2 deg diameter) was always shown at screen center. Each trial presented a pattern of white horizontal (non-target stimuli) or vertical

lines (target stimuli) (4.1 x 0.2 deg) randomly distributed across the screen, against a black background. Target stimuli appeared on a total of 30 trials (~3% of all trials) and they were distributed pseudo-randomly, every 26–40 trials; participants were asked to press the space bar whenever a target was detected. Moreover, a white verbal cue (0.88 deg text-height, e.g., "long hair", "jet lag") was placed under the fixation spot, in 192 trials (18.8%). The verbal cues were selected from the Italian adaptation of the original standardized pool of 800 word-phrases [22]. In our Italian adaptation study [45] we asked 10 independent judges to rate the level of familiarity, imageability and concreteness of the cues on a 7-point scale (1 = low and 7 = high). Equal numbers of neutral ($n = 64$), positive ($n = 64$), and negative ($n = 64$) cues were presented during the task.

**Thought questionnaire.** After completing the vigilance task, participants were asked to give details of their reported mental contents. For each content, they were asked to indicate: (i) the temporal focus, distinguishing between "past", "present", "future", and "atemporal", (ii) whether it was general or specific, (iii) the emotional valence of the thought on a 7-point scale (-3 = very unpleasant; 0 = neutral; +3 = very pleasant). Participants received instructions on how to distinguish the different temporal focus categories. Specifically, they were told that an "atemporal" mental content would refer to any thought with no specific temporal orientation (i.e., "I am very shy"), a "present" mental content would refer to any thought related to something occurring either here and now or in the current period of life, a "past" mental content would refer to any thought related to something occurred prior to begin the task (more or less remote), and a "future" mental content would refer to any thought related to something occurring after the end of the task (more or less distant in the future). Participants were also asked to rate on a 7-point scale their overall level of concentration (1 = not at all concentrated; 7 = fully concentrated) and boredom (1 = not at all; 7 = very bored) experienced during the vigilance task.

## Procedure

Participants were tested individually. When first welcomed into the laboratory, they were briefly introduced to the research project and eye-tracking recording technique, they were informed that they would take part in a study on concentration and its correlates, and they signed a consent form. Then, they received the instructions for the vigilance task: to detect target stimuli (vertical lines) among a large number of non-target stimuli (horizontal lines), by pressing the space-bar after each target stimulus, while keeping their gaze on the fixation point for the whole session. Participants were told that the computer would record automatically their response and the time needed to give it. We informed them of the occurrence of verbal cues in some of the trials, which were irrelevant to their task. To justify these, we provided a cover story (which was uncovered after the end of the experiment), that the experiment comprised two conditions tested in separate groups of participants, one testing how people focus on the patterns irrespectively of the verbal cues, and the other testing the opposite, focus on the cues irrespectively of the patterns. Finally, participants were told that the task was monotonous and that task-unrelated mental contents (e.g., thoughts, plans, considerations, past events, images, etc.) could pop into their mind spontaneously throughout the task. Any time this happened, they had to press a button on the keyboard (the letter L, marked with a white sticker) to interrupt the presentation and give a short description of the mental content; they would also have to indicate whether it was triggered by internal thoughts, an element in the environment, a cue-word on the screen (if so, they had to specify which word) or by no cue at all. These responses were recorded by the experimenter. This initial description should have been sufficient for them to identify the mental content at a later point in time, if necessary.

However, if the mental content was private and intimate, participants could label it as "personal" and provide only one relevant word instead of reporting a short description. Finally, participants were told that they should resume the task following the self-interruption and after providing a short description of their mental contents.

After the instructions, participants went through 20 practice trials and then proceeded to complete two sessions of 510 trials each. At the end of the sessions, subjects completed a questionnaire on their thought experience and we asked whether they had speculated about the actual aims of the study (if so, what they had thought) during the task and then they were debriefed and dismissed. The total session lasted approximately 100–120 min.

## Data encoding and analysis

All thoughts recorded by participants were read by two independent judges (first and third authors) and independently coded into three categories: the mental contents could be either task-related interferences (TRIs) or task-unrelated thoughts (TUTs); TUTs were further classified as either external distractions (EDs) or MW episodes. TRIs included all cases where a reference was made to task features or to the participant's overall performance (i.e., thoughts about the experiment's duration). EDs included all thoughts focused on task-unrelated sensations, either exteroceptive perceptions (i.e. a noise outside the room) or interoceptive sensations (i.e., bodily sensations). MW episodes included all thoughts that were unrelated to both the task and the sensory environment; however, these thoughts could be triggered by internal stimuli (e.g. other thoughts, feelings), external stimuli (including the verbal cues), or by no trigger, according to participants' reports. For both categorisations (TRIs *vs*. TUTs, and MW *vs*. EDs), we computed Kappa as inter-rater reliability between the coders and the inter-rater agreement resulted to be very good (*Kappa* = 0.96, *SE* = 0.03 and inter-rater agreement *Kappa* = 0.99, *SE* = 0.01, respectively). Minor disagreements were resolved by discussion.

Next, an off-line analysis examined the eye-tracking output and excluded time-points with unrealistic pupil-size recordings (i.e., values outside the 90th percentile of each 2 sec. long trial) and resampled the remaining time-points at 20Hz.

To perform the analyses on the pupillometry data, trials were sorted based on their timing relative to a cue later identified as "trigger" or "post-MW report" (i.e., 0, 1, 2 trials after the word, where the trial 0 was the one where the word was shown). We used as "baseline" pupil diameter the average diameter in the reference event (trial "0"). We studied the timecourse of pupil diameter over trials after subtracting this baseline. Since we were interested in the timecourse of MW episodes (how MW unfolds over time, as measured in seconds), we selected only MW events that met three criteria: the latency between the verbal cue and the self-interruption was at least 6 seconds (allowing us to see the unfolding of MW over three trials: the trigger-trial, 1-post and 2-post trigger); in this period, only non-target horizontal lines were shown (to avoid the confounding effects of cue-words and targets on pupil diameter) and reliable pupil recordings were collected. This window was only used for analyzing pupil traces, not for analyzing mind-wandering durations; the temporal window (and resulting selection criteria) used for pupillometry has absolutely no impact on the behavioral results (MW latencies).

Statistical analyses relied on a Linear-Mixed Model approach, motivated by the considerable sample size variability across subjects. In this approach, individual trials from all subjects are compared with a model comprising both the effect of experimental variables ("fixed effects") and the variability across participants ("random effects"). Random effects were coded by allowing subject-by-subject variations of the intercept of the model. In all cases, the dependent variable is "baseline corrected pupil diameter", which we obtained by averaging pupil diameter in a pre-specified temporal window of each trial and subtracting the average pupil

diameter in a reference temporal window. Please refer to the results section for specific defini-tions of the temporal windows for averaging and baseline-subtractions. We used standard MATLAB functions provided with the Statistics and Machine Learning Toolbox (R2015b, The MathWorks). Specifically, the function "fitlme(data, model)" fits the linear-mixed model to the data, yielding an object "lme" with associated method "ANOVA" that returns $F$ statistics with associated degrees of freedom, and $P$ values for each of the fixed effect terms and "CoefT-est" for post-hoc comparisons.

Data available at https://doi.org/10.5281/zenodo.3673044

## Results

### Performance on vigilance task

Performance on the vigilance task was near-perfect for all participants. Out of 30 targets, there were 0.46 ($SD$ = 0.66) misses and 0.17 ($SD$ = 0.38) false alarms. The mean reaction time associ-ated with correct detections was 767.13 msec. ($SD$ = 124.42 msec.). The mean level of concen-tration experienced during the task was 4.92 ($SD$ = 1.06) out of 7 and the mean level of boredom was 3.04 ($SD$ = 1.57) out of 7.

### Frequency and properties of MW reports

Participants reported a total of 400 mental contents ($M$ = 16.67, $SD$ = 16.83, range 1–63). Out of the all mental contents, 27 reports (6.75%) were classed as TRI reports ($M$ = 1.13, $SD$ = 1.54, range 0–6), and 373 reports (93.25%) were classed as TUT reports ($M$ = 15.54, $SD$ = 16.59, range 1–61). Out of all the TUTs, 40 reports (10.72%) were classed as external distractions ($M$ = 1.67, $SD$ = 1.93, range 0–6) and 333 reports (89.28%) were classed as MW ($M$ = 13.88, $SD$ = 15.48, range 0–58). One out of 333 MW reports was excluded from the analysis of pupil diameter because of inaccurate recording of time of MW interruption and report.

Out of 333 MW episodes, 67.57% were reported as triggered by a verbal cue previously shown on-screen, 3.90%, by internal thoughts, 4.50% by environmental triggers and 18.32% by no trigger. The remaining 19 MW episodes were indicated by participants as elicited by multi-ple cue-words (i.e., more than one specific cue-word; n = 7), by unknown cue-word(s) (i.e., participants reported that they did not remember which cue-word triggered the MW episode; n = 8), or by multiple triggers (e.g., participants reported that their thoughts were elicited by both a cue-word presented onscreen and a sound occurring outside the room; n = 4).

Finally, out of all the MW episodes triggered by the verbal cues, 26.67% were triggered by neutral cue-words, 35.11% by positive cue-words and 38.22% by negative cue-words. Thus, the vast majority (73.33%) of these reports were elicited by verbal cues with emotional valence.

At the end of the vigilance task, participants coded each of their reported thoughts as past-oriented, present-oriented, future-oriented or atemporal thoughts. Out of the all MW epi-sodes, 133 episodes (39.94%) were classed as past-oriented thoughts, 21 episodes (6.31%) were classed as present-oriented thoughts, 53 episodes (15.91%) were classed as future-oriented thoughts, and 126 episodes (37.84%) were classed as atemporal thoughts. Moreover, out of all the MW episodes, 167 episodes (50.15%) were classed as specific, whereas 166 episodes (49.85%) were classed as general. Finally, 96 episodes (28.83%) were rated as neutral, 118 epi-sodes (35.43%) were rated as negative, and 119 (35.74%) were rated as positive.

### Latency data

For each MW episode indicated by participants as triggered by a cue-word, we computed the time-interval between a verbal cue and the task-interruption where a MW episode was

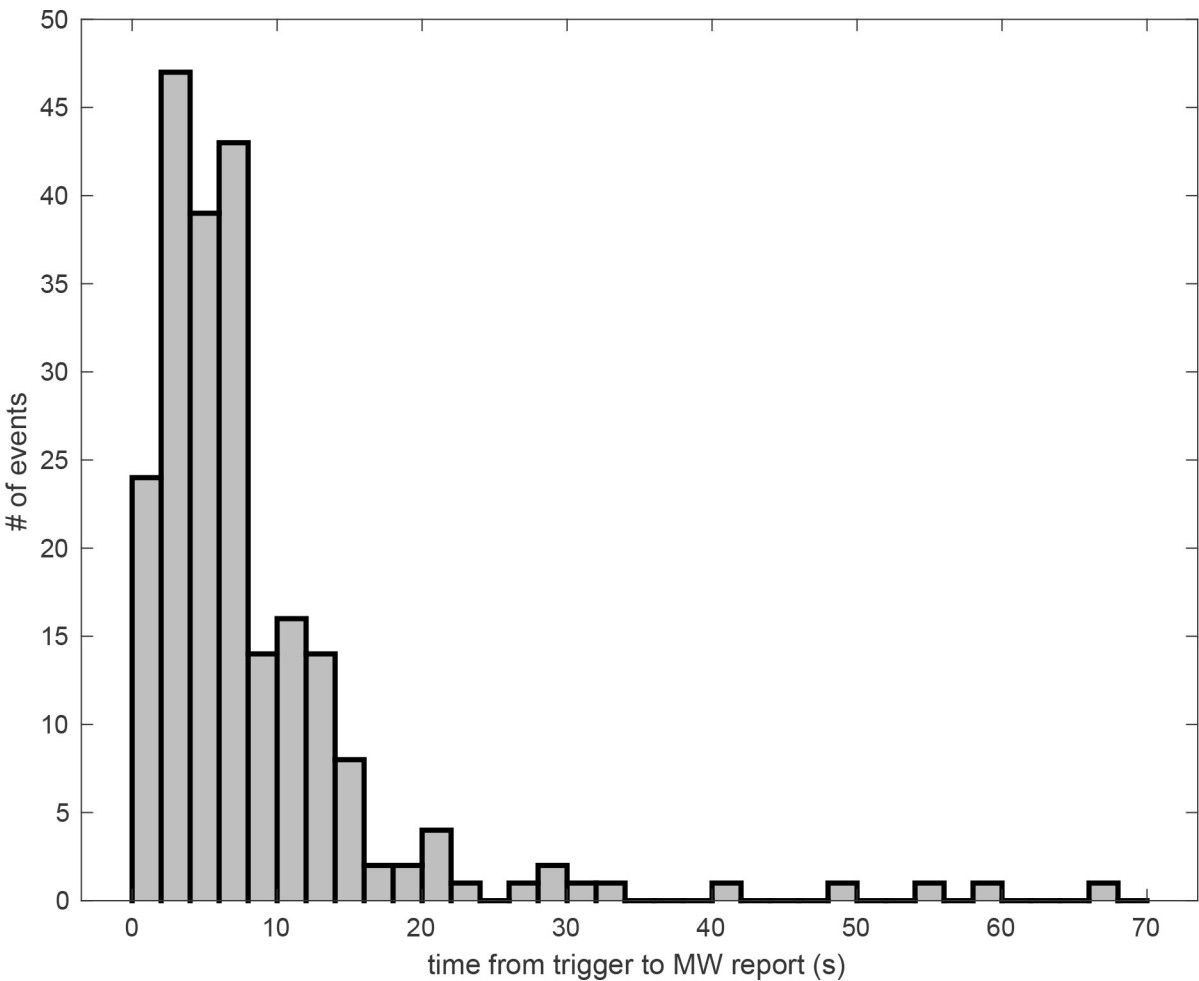

**Fig 1. Distribution of latency data.** Distribution of the time-intervals between a verbal cue and a MW report.

reported as triggered by said verbal cue (see also studies on involuntary memories for a similar procedure to obtain retrieval times; e.g., 22, 43, 45).

The median latency of the 225 MW episodes triggered by a verbal cue was 6.19 sec and the mean latency was 8.54 s (*SD* = 9.93), which can be interpreted as the average duration of the MW episode (from its onset, marked by the trigger presentation, to its end, marked by the self-interruption of the task). The distribution of latency data is reported in Fig 1.

We analyzed whether and how the phenomenological characteristics of MW episodes (temporal focus, specificity and emotional valence) affected their durations by means of a Bayesian factor analysis of a Linear-Mixed Model. Specifically, we analyzed the effect of temporal focus of MW (with four levels) with a model including the random effect of the variable "subject". The base-10 logarithm of the Bayes Factors (lg10BF) associated with the model is -1.38 ($<$-0.5, which implies robust evidence in support of the null hypothesis, that the model is not a good representation of the data). Similarly, we evaluated a linear-mixed model representing the effects of emotional value (with three levels) and specificity (with two levels), again with subjects as random effect; the model was associated with a lg10BF of 4.22, implying robust evidence that this model is also not a good representation of the data.

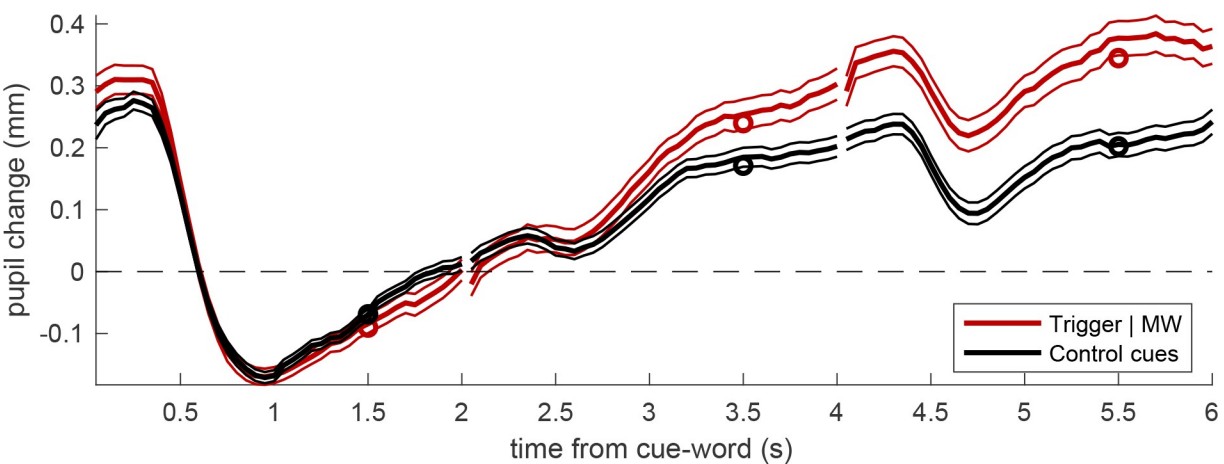

**Fig 2. Pupil traces aligned to the average pupil diameter during the presentation of the verbal cue.** Thick lines give the average across all trials, thin lines show the s.e. and circles show the average values entered in the LMM analysis (average over the second half of each trial). The two traces give the average pupil diameter following a verbal cue that was later reported as triggering a MW episode, or a verbal cue presented during a supposedly on-task period, i.e. the period immediately following the resuming of the task after a self-caught interruption and MW report.

### Pupillometric correlates of mind wandering

In the first analysis of pupillometry data, we analysed the timecourse of pupil diameter over two trials after a trigger (a verbal cue presented before a MW event and identified by the participant as the trigger of said event, n = 107) or a non-trigger (a verbal cue presented after a MW report, hence unlikely to be trigger of further distractions, reported or non-reported, n = 226; the mean temporal distance -in seconds- between the MW report and the following cue-word was 8.50 sec., SD = 3.22 sec).

Fig 2 shows the traces, aligned to the average pupil diameter during the verbal cue presentation (baseline window defined as the first half of the verbal-cue trial). Verbal cues (white text on black background) evoked strong pupillary constriction (the pupillary light response), which recovers over several seconds producing a progressive pupil dilation. This pupil dilation is more pronounced for trigger cues than non-trigger cues.

In order to statistically assess this effect, we summarized pupil traces by taking the average pupil diameter in the last second of each trial (the farthest from the baseline window). This choice is based on the assumption that the effect of MW develops gradually over time.

These values were entered in a Linear-Mixed Model analysis, with two fixed-factors: type of cue-word (trigger and non-trigger) and time from the cue (coded as number of trials), plus the random effect of subjects modelled as a variable intercept of the model. This revealed a significant interaction between the two fixed factors ($F(1,995) = 23.49$, $p < 0.00001$), which we further analysed with a series of post-hoc tests. These showed that cue-type has a significant effect on both trials following the cue, the first ($F(1,331) = 9.15$, $p = 0.00268$) and the second ($F(1,331) = 22.42$, $p < 0.00001$).

These findings were confirmed when we used as control-cues a subset of word-cues with emotional content (N = 156)–compared against the same N = 107 trigger-cues which, for the most part, had emotional content. The Linear-Mixed model analysis revealed a significant interaction between type of verbal cue (trigger, and emotional non-trigger) and time from the cue ($F(1,785) = 19.01$, $p = 0.00001$), and again indicated a large effect of cue-type on both trials following the cue, the first ($F(1,261) = 10.39$, $p = 0.00143$) and the second ($F(1,261) = 20.00$, $p = 0.00001$).

The analyses reported so far used stringent selection criteria, which allowed for comparing pupil diameter over trials for which a continuous pupil recording was available (imposing equal numerosity at the three delays) and was not contaminated by extraneous factors (such as the presentation of other cues or target stimuli in the time-window between the trigger-cue and the self-interruption). However, these specific requirements reduced the amount of MW episodes (to less than half of the total sample), raising concerns about the representativeness of our pupillometric results. To address these concerns, we ran a second analysis in which all MW events were included, and no criteria of exclusion were applied. This analysis included a variable number of trials which depends on the variable latency of MW episodes. Nevertheless, the LMM analysis revealed the same significant effects as found in our main (restrictive) analysis: significant interaction of the two fixed factors, type of cue and time from cue ($F(1,1336) = 34.32$, $p < 0.000001$); significant post-hoc effect of cue-type on both the first and the second trial after the cue ($F(1,428) = 23.92$, $p < 0.000001$) and the second ($F(1, 430) = 39.72$, $p < 0.000001$).

A second technical issue concerns the selection of control-cues. These were defined as the first cue-words that followed a self-interruption. However, their temporal distance (in seconds) from the self-interruption was variable and in some cases relatively large ($>15$ seconds), raising the possibility that participants may have relapsed into a MW state by the time the cue-word was presented. We excluded these cases in a further control analysis, in which control-cues were presented 6 seconds or less after the self-interruption ($n = 90$; mean distance: 5.18 sec; SD = 0.99 sec), and still obtained the same results as in the main analysis: significant interaction between the factors trial type and time ($F(1,587) = 15.88$, $p = 0.00008$); significant effect of trial type on the first ($F(1,195) = 6.16$, $p < 0.05$) and the second trial following the cue ($F(1,195) = 14.87$, $p < 0.0005$) and even a significant effect of trial time on the cue-trial ($F (1, 195) = 5.28$, $p < 0.05$).

Finally, following [13], we complemented these analyses of pupil traces aligned to cue-words, with a second series of analyses, with pupil traces aligned to the MW reports (Fig 3)–or to "control trials" (5 trials following every self-interruption).

Pupil diameter preceding these control trials can also be seen as the timecourse that pupil diameter should regularly show in this experiment, without the intrusion of a mind wandering episode. Also, in this case, there was a relative pupil dilation in the trials preceding the MW report. To show this statistically, we aligned traces to the average pupil diameter on the last

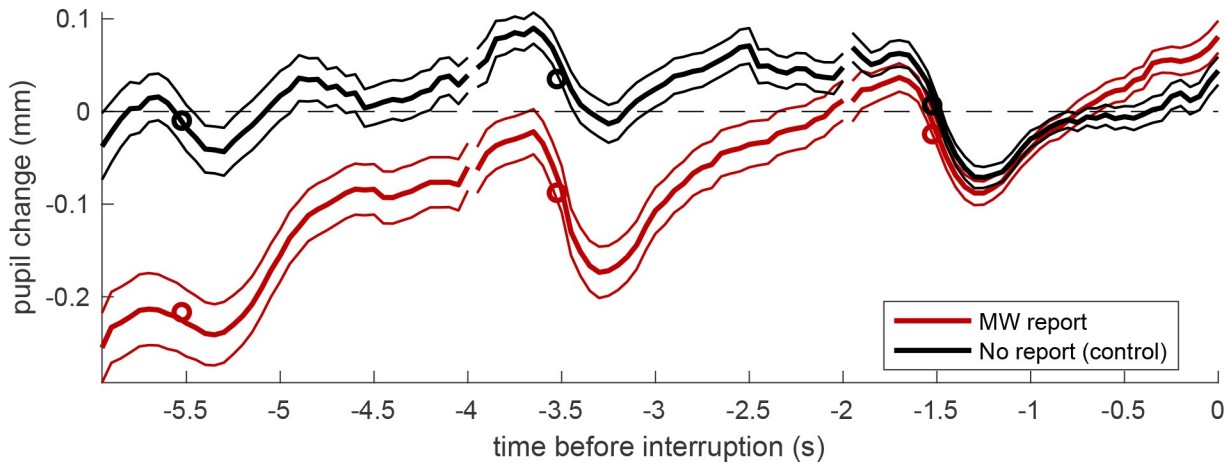

**Fig 3. Pupil traces aligned to the average pupil diameter on the last trial before the MW report or the control trial.** Thick lines give the average across all trials, thin lines show the s.e. and circles show the average values entered the LMM analysis (average over the second half of each trial).

trial before the MW report or the control trial, then assessed pupil diameter on each trial as the mean pupil diameter in the first second of the trial (the farthest from the reference). In this analysis we followed the same restrictive criteria used to investigate the dynamics of MW triggered by a cue-word, comparing pupil diameter over trials for which a continuous pupil recording is available (imposing equal numerosity at the three delays). These criteria left 89 MW reports and 102 control trials. The LMM analysis revealed a significant interaction between the fixed-factors type of trial (MW/control) and time ($F(1,569) = 18.23$, $p = 0.00002$). At three trials preceding the reference trial, pupil diameter leading to a MW report could be clearly differentiated from the usual pupil diameter during the experiment ($F(1,189) = 25.48$, $p < 0.00001$). This confirms that there is pupil dilation leading up to a MW report.

We also ran a control analysis including all MW events, independent of their duration and of the intervening stimuli (e.g. vertical lines targets). This yielded 312 trials at delay 0 (all trials for which a reliably pupil recording could be obtained), 308 at delay -1 and 303 at delay -2 (the number reducing due to the variable duration of MW episodes) and we still found significant type of episode x time interaction ($F (1,1762) = 8.72$, $p < 0.01$) and significant post-hoc effects of episode type at trial -3 ($F (1, 568) = 18.41$, $p < 0.00005$), -2 ($F (1,588) = 23.18$, $p < 0.000001$) and -1 ($F (1,604) = 9.19$, $p < 0.005$).

## Discussion

Several recent studies have shown that, in the majority of cases, MW is a cue-dependent phenomenon, triggered by both internal or external events. Specifically, there is evidence that both task-irrelevant and task-relevant external stimuli [12–20] might act as triggers for MW episodes. In a recent study, Pelagatti et al. [13] found a significantly larger pupil dilation following cue-words reported by participants as the trigger of MW compared to non-trigger words (with similar emotional content), and the pupil dilation appeared to increase over time. In the study by Pelagatti et al. [13] MW was assessed by using the probe-catching method. In the present study, we globally replicated these findings using the self-catching method, which "*provides a straightforward assessment of the number of mind wandering episodes that reached meta-awareness*" [28, p.322], thereby allowing investigation of the pupil correlates of MW that an individual becomes aware of (aware MW).

Our results on the increased pupil diameter during MW are in line with some previous findings [e.g. 13, 49; see also 50, for larger pupil diameters in participants who retrospectively reported more MW compared to those who reported less MW]. However, other studies have reported the opposite finding: reduced pupil dilation during MW [e.g. 23, 35, 51,52]. The control of pupil diameter is affected by a variety of factors [e.g. 53–62], and ultimately reflects the balance between the parasympathetic and sympathetic systems [reflecting the level of arousal, e.g. as linked with cognitive and emotional load, see 26, 60, 63]. When light level is constant, we can expect pupil dilation every time the cognitive or emotional load is increased [e.g., 26]. However, given any two tasks, it is not always obvious which of the two is associated with higher load. In the MW literature, pupil diameter is compared between two conditions: epochs when the participant is focused on the task, and seldom interspersed with mind-wandering events. Are the latter events associated with more or less load than on-task epochs? And, consequently, should we expect pupil dilation or constriction during a MW episode? We believe that the answer will strongly depend on two main related factors, namely (i) the operational definition of MW and its characteristics (ii) the characteristics of the main task. In this sense, we agree with Konishi et al. [52; see also 51, 64], suggesting that the content of MW and the context in which it emerges are determinant for its relation to other neurocognitive variables (including pupil size).

As for the first aspect, we believe that one should not lump together different attentional states with disparate dynamics and different levels of cognitive load. This would make it impossible, a priori, to predict the direction of pupil modulations. In the present study, we analyzed the mental contents reported by participants and distinguished episodes of MW from external distractions (including bodily/physical sensations) and task-related interferences.

The MW episodes triggered by external stimuli could be distinguished from external distractions, as participants were not focusing their attention on the external stimuli, but they were thinking about private thoughts and feelings, including autobiographical memories/ future plans and simulations, that were prompted by an external stimulus. On the contrary external distractions referred to "thoughts pertaining to the immediate environment decoupled from the task" [14, p. 11].

Following this categorization, we found that the mental contents during MW episodes were relatively complex personal thoughts, which consisted of personal projections into the personal past (i.e., autobiographical memories) and the future (e.g., future planning, upcoming personal events)–certainly more complex than the mental content accompanying the detection of simple visual stimuli required by our vigilance task.

The other factor is the nature of the main task (which is interrupted by the MW episodes), and specifically the perceptual and cognitive load associated with it. In our case the presence of pupil dilation during MW episodes may be interpreted considering the monotonous and undemanding main task (more repetitive and less challenging than other tasks used in previous research on pupillometry and mw) and the presence of cue-words, which might have charged the latter with particularly high emotional/cognitive load. The fact that different predictions about the pupil correlates of MW might be advanced depending on the operationalization of MW and the characteristics of the main task does not reduce the importance of pupillometry, but it emphasizes the necessity of a clear definition of the relevant parameters.

In the research field on MW, both self-caught and probe-caught methods have been recognized as valid approaches although, as we reviewed above, they might be considered distinct measures of MW, somehow tapping into relatively different kinds of MW. In some regards, the less popular self-caught method might be considered more ecological than the probe-caught, because it is more similar to the way we become aware of our shifts of attention during task in daily-life [27]. Moreover, as pointed out by Faber and D'Mello [14] another advantage of the self-caught method is that MW "*reports can occur at any time, independent of whether and when a participant received a thought probe*" [14, p. 9]. However, with probe-caught method we can tap into both aware and unaware MW episodes, making a more comprehensive evaluation of the experience of MW.

In the present study, we used the self-caught method for two reasons: first, to verify whether self-caught MW showed the same kind of pupillometry effects found in our previous study on probe-caught MW [13] and, second, to extend our investigation of the temporal dynamics of MW, including an estimate of the latency of MW episodes (see next paragraph). As for the first aspect, here we could replicate, with a more homogeneous subset of MW episodes (only aware MW), the patterns found with the probe-caught MW (including both aware and unaware MW episodes). However, since we did not combine self-caught and probe-caught method in the same task, we could not directly compare aware and unaware MW episodes, to verify whether and how the level of meta-awareness associated with different MW episodes modulate pupillometry correlates. Future studies are needed to systematically investigate this aspect, by combining the two methods within the same paradigm.

The presence of cue-words gave us the opportunity to estimate the latency of MW episodes that these had triggered, by computing the time-interval between the presentation of triggering

cue-words and the self-reports of MW. The latency of a MW episode could be considered as the time needed for the arising of thought and for becoming aware of it in order to report it. We found high variability in the duration of MW, with 40% of relatively short MW episodes, lasting 5 sec or less. These findings are in line with the ones reported by Klinger [34], who trained participants to estimate the duration of their thoughts and found a median estimate of thought segment duration of 5 sec. The large variability of our latency estimates, however, raises a methodological flag for several previous studies, which used fixed time-windows to compare on-task and MW episodes (and very variable time-windows across studies [e.g., 10 sec. in 49, 65; 6.5 sec. in 66; 3 to 8 sec. in 67; 5 sec. in 68, 69; 4.8 sec. in 70]). By using a fixed time-window, identical for all episodes and participants, these studies have implicitly assumed that the duration of MW episodes is relatively constant. However, both our results and the ones reported by Klinger [34] clearly show that the duration of MW is highly variable, within and across subjects. These findings should stimulate research into the factors that might explain and predict this variability and help our understanding of the processes that lead to the variable timing of MW.

In conclusion, classic experiments on MW have left many of its key properties, and particularly its temporal properties, unexplored. Future research will need new objective tools to clarify these aspects; we submit that our pupillometry paradigm can contribute to this quest, providing an objective and reliable marker of the onset and temporal unfolding of MW episodes.

## Author Contributions

**Conceptualization:** Manila Vannucci.

**Data curation:** Claudia Pelagatti.

**Formal analysis:** Paola Binda.

**Funding acquisition:** Paola Binda.

**Investigation:** Claudia Pelagatti.

**Methodology:** Claudia Pelagatti, Paola Binda, Manila Vannucci.

**Supervision:** Manila Vannucci.

**Writing – original draft:** Manila Vannucci.

**Writing – review & editing:** Paola Binda, Manila Vannucci.

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
