## [Decision Letter · Decision Letter 0]

10 Jan 2020

PONE-D-19-33603

A closer look at the timecourse of mind wandering: pupillary responses and behavior

PLOS ONE

Dear Dr. Vannucci,

Thank you for submitting your manuscript to PLOS ONE. After careful consideration, we feel that it has merit but does not fully meet PLOS ONE’s publication criteria as it currently stands. Therefore, we invite you to submit a revised version of the manuscript that addresses the points raised during the review process.

As you can see below, both reviewers positively evaluate your manuscript. However, each raises a number of points that need to be addressed. In particular, reviewer 1 raises excellent points with regard to the description of the methods, procedure, and analyses. Given that one of PLOS ONE’s publication criteria is that experiments must be described in sufficient detail, it is necessary to add these details and justifications. I also agree with reviewer 1 that it is necessary to look at the effect of longer latencies (main point 8), and to quantify whether there is really no effect of the content of mind wandering (main point 9). I agree with both reviewers that the discussion needs to comprehensively discuss the effects of self- and probe-caught mind wandering, as well as to be more specific about the task-specific aspects of the relationship between mind wandering and pupil diameter (reviewer 2, point 4). I also agree with both reviewers that it would be helpful to have a clearer definition of mind wandering and a more comprehensive overview of differences and similarities between outcomes of self-caught and probe-caught mind wandering studies to motivate the current study. I would therefore like to invite a revision that addresses these points.  

We would appreciate receiving your revised manuscript by Feb 24 2020 11:59PM. To enhance the reproducibility of your results, we recommend that if applicable you deposit your laboratory protocols in protocols.io, where a protocol can be assigned its own identifier (DOI) such that it can be cited independently in the future. For instructions see: http://journals.plos.org/plosone/s/submission-guidelines#loc-laboratory-protocols

We look forward to receiving your revised manuscript.

Kind regards,

Myrthe Faber

Academic Editor

PLOS ONE

Journal Requirements:

https://www.journalofcognition.org/articles/10.5334/joc.41/

In your revision ensure you cite all your sources (including your own works), and quote or rephrase any duplicated text outside the methods section. Further consideration is dependent on these concerns being addressed.

3. We note that you have stated that you will provide repository information for your data at acceptance.

Should your manuscript be accepted for publication, we will hold it until you provide the relevant accession numbers or DOIs necessary to access your data.

If you wish to make changes to your Data Availability statement, please describe these changes in your cover letter and we will update your Data Availability statement to reflect the information you provide.

Reviewers' comments:

Reviewer's Responses to Questions

**Comments to the Author**

1. Is the manuscript technically sound, and do the data support the conclusions?

Reviewer #1: Yes

Reviewer #2: Yes

2. Has the statistical analysis been performed appropriately and rigorously? 

Reviewer #1: Yes

Reviewer #2: Yes

3. Have the authors made all data underlying the findings in their manuscript fully available?

Reviewer #1: No

Reviewer #2: Yes

4. Is the manuscript presented in an intelligible fashion and written in standard English?

Reviewer #1: Yes

Reviewer #2: Yes

5. Review Comments to the Author

Reviewer #1: summary:

This manuscript looks at the dynamics of mind-wandering in quite a smart way: by using a task in which cues are presented that can trigger mind-wandering. Participants were asked to do a boring vigilance task, in which cue words were embedded. Whenever they noticed their mind had wandered, they we asked to press a button and report on this mind-wandering. The researchers investigated the lag between cue-words to which participants attributed their mind-wandering and the self-caught mind-wandering responses. They also examined pupil size and found it was significantly larger for mind-wandering compared to control trials.

main points

Overall I think this is a very nice study with interesting methods and results. Nevertheless, I have some questions about the methods and results.

1. The cue phrases play a major role in this method. Naively thinking, it would seem to me that the nature of these phrases makes a huge difference. How were these phrases chosen? (this would be helpful to discuss in the introduction on p.4)]

2. It would be helpful to synthesize the review in the introduction the findings regarding differences and similarities between the outcomes of self-caught and probe-caught mind-wandering studies and generate hypotheses for the current study

3. There is quite a substantial number of participants "not complying with task instructions." What do you mean with that? And is this in any way related to your reliance on self-caught thought probes? Also in relation to quality of participant's report: have you analyzed the relation between ratings of concentration and boredom and task performance? Could this be used to weed out participants who have unreliable thought probe responses?

4. What is missing from the task instructions is how participants were briefed on speed vs. accuracy. This could have substantial influence on both task performance and the tendency to mind-wander.

5. Why did you define mind-wandering episodes as 0-2 trials after the trigger or 1-3 trials before the self-report? Doesn't this specific requirement lead to specific results on the mind-wandering durations?

6. What proportion of mind-wandering episodes was removed by the three criteria specified on p.10?

7. The analysis of the content of mind-wandering episodes is very interesting. How does that compare to your own and other studies that were using probe-caught responses, and that used "natural mind-wandering" rather than cue-evoked mind-wandering? Did you examine whether past-related mind-wandering was mostly negative as previously reported by e.g., Jonathan Smallwood?

8. I am concerned about your removal of the latencies larger than 2.5 SD since this assumes that these latencies are normally distributed, and most likely they are not (the graph looks more exponential). What happens if you do not remove these longer latencies? They may in fact carry a lot of relevant information!

9. In the latency data, no significant effects of content of mind-wandering were found. This could arise fron two causes: there really no being a difference, or alternatively, the data being too uncertain. To determine which one, one can use a Bayes Factor, which can easily be done with the BayesFactor in R for LMEs.

10. Why did you use the last second from each trial in the LMEs? Why is it important to be far away from the baseline interval?

11. the discussion lacks a section comparing self-caught and probe-caught mind-wandering. How do the results compare in this study, as well as previous studies in the literature?

12. mind-wandering tends to increase during the course of the task. Was there an increase in self-caught mind-wandering responses during the second part of the task? If not, maybe this would reflect a difference between cue-evoked mind-wandering and more "natural" mind-wandering.

13. why are the data only available upon acceptance? Would be good to make these available during the review phase

minor points

1. p.4: Is there a reason why people think that verbal cues may increase the proportion of past-related mind-wandering reports?

2. p.7 How is the pupil diameter transformed from pixels to milliseconds using this artificial pupil? Do you have a reference for that?

3. p.11 "One out of 133 MW reports was excluded from the analysis of pupil diameter because of inaccurate recording of time of MNW interuption and report." How is this possible given the fact that the task is administered by computer?

4. On p.14: "or a non-trigger (a verbal cue presented after a MW report)". How long after a MW report were such cues presented and can we be sure that participants were really not mind-wandering anymore?

Reviewer #2: The authors investigate changes in pupil diameter associated with self-caught mind wandering using a relatively new vigilance task that included intermixed task-irrelevant cue-phrases that, according to results, frequently prompted mind wandering. An interesting aspect of this working offers insights into the temporal unfolding of MW, showing that a self-report occurred, on average, about 7 s following a cue-phrase that reportedly triggered the MW episode. This time period corresponded to increases in pupil dilatation, which replicates past findings using a probe-caught method to measure mind wandering.

For the most part the paper is well written (however, see point 3 below). Although the work does not necessarily contribute novel insights, given that it is just a conceptual replication of the authors’ previous findings, the methodological approach is sound. I have a few minor suggestions for the manuscript itself that could be easily addressed in a revision.

1.) A growing body of research has discussed the temporal dynamics of mind wandering with much theoretical debate. Given that the authors operationalize MW here as task-unrelated thought but attempt to speak to the temporal unfolding of mind wandering, discussion about the Dynamic Framework of mind wandering should be included.

Christoff, K., Irving, Z. C., Fox, K. C., Spreng, R. N., & Andrews-Hanna, J. R. (2016). Mind-wandering as spontaneous thought: a dynamic framework. Nature Reviews Neuroscience, 17(11), 718.

Christoff, K., Mills, C., Andrews-Hanna, J. R., Irving, Z. C., Thompson, E., Fox, K. C., & Kam, J. W. (2018). Mind-wandering as a scientific concept: cutting through the definitional haze. Trends in cognitive sciences, 22(11), 957-959.

Seli, P., Kane, M. J., Smallwood, J., Schacter, D. L., Maillet, D., Schooler, J. W., & Smilek, D. (2018). Mind-wandering as a natural kind: A family-resemblances view. Trends in cognitive sciences, 22(6), 479-490.

Seli, P., Kane, M. J., Metzinger, T., Smallwood, J., Schacter, D. L., Maillet, D., Schooler, J. W., & Smilek, D. (2018). The family-resemblances framework for mind-wandering remains well clad. Trends in cognitive sciences, 22(11), 959-961.

2.) The authors are inconsistent with how they describe MW. Sometimes MW is described as spontaneous and sometimes as being cue-dependent, perhaps because of the current state of the field. For instance, in the Introduction, the authors quote Klinger “spontaneous thoughts are probably triggered by cues…,” which in itself is a bit of a paradox: how could something be both spontaneous and also triggered? In the first paragraph of the Introduction, the authors write that “our attention drifts away… which mainly occurs spontaneously” but then point to research challenging this traditional view. It might be better, then, to describe mind wandering as “traditionally thought to drift” and/or “traditionally thought to occur spontaneously” before describing previous literature to the contrary. Similarly, in the first sentence of the Discussion, the authors write that “recent studies have shown that MW is a cue-dependent phenomenon,” which sounds like MW is always triggered by some sort of cue, but then the next sentence says that external and internal stimuli “might” act as triggers. Although a minor point, it would be helpful for the readers, especially for the broad audience of PLOS ONE, if the authors provide consistency throughout the piece.

3.) Only two decimal places are needed for reported statistics. The authors are inconsistent with this throughout the Results section.

4.) The second paragraph of the Discussion is poorly written and difficult to follow. I think the task-specific aspects of the MW-pupil diameter relationship could be described with more precision. I also do not agree that “it is impossible” to predict the direction of pupil modulations in a new experimental paradigm. One could look for similarities across task parameters to make predictions. Is this why the authors do not provide a justification for sample size? One should be included.

5.) The following article is highly relevant to the current work:

Faber, M., & D’Mello, S. K. (2018). How the stimulus influences mind wandering in semantically rich task contexts. Cognitive research: principles and implications, 3(1), 35.

6. PLOS authors have the option to publish the peer review history of their article (what does this mean?). If published, this will include your full peer review and any attached files.

Reviewer #1: No

Reviewer #2: No

---

## [Author Response · Author response to Decision Letter 0]

9 Mar 2020

Dear Editor,

We would like to thank you and the reviewers for your helpful suggestions on the manuscript “A closer look at the timecourse of mind wandering: pupillary responses and behaviour” for your consideration for publication as Research Article in PLoS One.

We greatly appreciate the constructive and thoughtful comments we received from you and from the reviewers, who made many valuable points. 

We are pleased to submit to your attention the revised version of our manuscript. In the current version we provide detailed responses to reviewers’ concerns and we made every effort to respond to the reviewers points. 

Our responses to reviewers' comments (including figures) are reported in both the "Cover letter" and the "Responses to reviewers" word files.

Best regards

Manila Vannucci

---

## [Decision Letter · Decision Letter 1]

13 Apr 2020

A closer look at the timecourse of mind wandering: pupillary responses and behavior

PONE-D-19-33603R1

Dear Dr. Vannucci,

We are pleased to inform you that your manuscript has been judged scientifically suitable for publication and will be formally accepted for publication once it complies with all outstanding technical requirements. Reviewer 1 also makes a few textual suggestions that you might want to incorporate. 

With kind regards,

Myrthe Faber

Academic Editor

PLOS ONE

Additional Editor Comments (optional):

Reviewers' comments:

Reviewer's Responses to Questions

**Comments to the Author**

1. If the authors have adequately addressed your comments raised in a previous round of review and you feel that this manuscript is now acceptable for publication, you may indicate that here to bypass the “Comments to the Author” section, enter your conflict of interest statement in the “Confidential to Editor” section, and submit your "Accept" recommendation.

Reviewer #1: All comments have been addressed

Reviewer #2: All comments have been addressed

2. Is the manuscript technically sound, and do the data support the conclusions?

Reviewer #1: Yes

Reviewer #2: Yes

3. Has the statistical analysis been performed appropriately and rigorously? 

Reviewer #1: Yes

Reviewer #2: Yes

4. Have the authors made all data underlying the findings in their manuscript fully available?

Reviewer #1: Yes

Reviewer #2: Yes

5. Is the manuscript presented in an intelligible fashion and written in standard English?

Reviewer #1: Yes

Reviewer #2: Yes

6. Review Comments to the Author

Reviewer #1: Overall, the paper has improved a lot. Thanks for taking my comments so seriously! I also hope you are doing OK in the current dramatic situation...

Some minor comments that may make the paper better:

p.15 Mean level of concentration/mean level of boredom: it would be helpful to mention the full range of the scale, e.g., was 4.92 out of 7 (just because the reader may have forgotten...)

p.16 Bayesian factor -> Bayes Factor (BF)

I would suggest indicating why you chose the specific analysis windows (which you indicated in response to my point 10). I suspect this would be helpful to many readers

It would be a good idea to mention the location of the data in the methods section.

Reviewer #2: (No Response)

7. PLOS authors have the option to publish the peer review history of their article (what does this mean?). If published, this will include your full peer review and any attached files.

Reviewer #1: Yes: Marieke van Vugt

Reviewer #2: No

---

## [Editor Report · Acceptance letter]

17 Apr 2020

PONE-D-19-33603R1 

A closer look at the timecourse of mind wandering: pupillary responses and behavior 

Dear Dr. Vannucci:

I am pleased to inform you that your manuscript has been deemed suitable for publication in PLOS ONE. Congratulations! Your manuscript is now with our production department. 

With kind regards,

on behalf of

Dr. Myrthe Faber 

Academic Editor

PLOS ONE